# Protein Biocargo and Anti-Inflammatory Effect of Tomato Fruit-Derived Nanovesicles Separated by Density Gradient Ultracentrifugation and Loaded with Curcumin

**DOI:** 10.3390/pharmaceutics15020333

**Published:** 2023-01-19

**Authors:** Ramila Mammadova, Serena Maggio, Immacolata Fiume, Ramesh Bokka, Maneea Moubarak, Gabriella Gellén, Gitta Schlosser, Giorgia Adamo, Antonella Bongiovanni, Francesco Trepiccione, Michele Guescini, Gabriella Pocsfalvi

**Affiliations:** 1Extracellular Vesicles and Mass Spectrometry Laboratory, Institute of Biosciences and BioResources, National Research Council of Italy, 80131 Naples, Italy; 2Department of Biomolecular Sciences, University of Urbino Carlo Bo, 61029 Urbino, Italy; 3MTA-ELTE Lendület Ion Mobility Mass Spectrometry Research Group, Institute of Chemistry, ELTE Eötvös Loránd University, H-1117 Budapest, Hungary; 4Institute for Biomedical Research and Innovation, National Research Council of Italy, 90146 Palermo, Italy; 5Department of Translational Medical Sciences, University of Campania Luigi Vanvitelli, 80138 Naples, Italy

**Keywords:** tomato, plant-derived nanovesicles (PDNVs), anti-inflammatory activity, in vitro analysis, loading methods, proteomics, curcumin, protein functional analysis

## Abstract

Plant-derived nanovesicles (PDNVs) have become attractive alternatives to mammalian cell-derived extracellular vesicles (EVs) both as therapeutic approaches and drug-delivery vehicles. In this study, we isolated tomato fruit-derived NVs and separated them by the iodixanol density gradient ultracentrifugation (DGUC) into twelve fractions. Three visible bands were observed at densities 1.064 ± 0.007 g/mL, 1.103 ± 0.006 g/mL and 1.122 ± 0.012 g/mL. Crude tomato PDNVs and DGUC fractions were characterized by particle size-distribution, concentration, lipid and protein contents as well as protein composition using mass spectrometry-based proteomics. Cytotoxicity and anti-inflammatory activity of the DGUC fractions associated to these bands were assessed in the lipopolysaccharide (LPS)-stimulated human monocytic THP-1 cell culture. The middle and the low-density visible DGUC fractions of tomato PDNVs showed a significant reduction in LPS-induced inflammatory IL-1β cytokine mRNA production. Functional analysis of proteins identified in these fractions reveals the presence of 14-3-3 proteins, endoplasmic reticulum luminal binding proteins and GTP binding proteins associated to gene ontology (GO) term GO:0050794 and the regulation of several cellular processes including inflammation. The most abundant middle-density DGUC fraction was loaded with curcumin using direct loading, sonication and extrusion methods and anti-inflammatory activity was compared. The highest entrapment efficiency and drug loading capacity was obtained by direct loading. Curcumin loaded by sonication increased the basal anti-inflammatory activity of tomato PDNVs.

## 1. Introduction

In recent years, plant-derived nanovesicles (PDNVs) have emerged due to their perspective towards biomedical and nanotechnological applications. They have been isolated from many homogenized plants and fruit juices, including tomato [1,2], citrus species [3,4,5], strawberry [6,7] and ginseng [8,9], just to mention some. In this paper, we use PDNVs as an umbrella term to refer to all kind of nano-sized particles, nanolipids, exosomes, nanovectors, etc. [10] isolated from plant organs containing different tissues after homogenization. Physical, morphological and molecular characterization showed that PDNV isolates contain membrane-enclosed structures that resemble mammalian extracellular vesicles (EVs). Moreover, PDNVs are efficiently taken up by mammalian cells through various mechanisms and possess bioactivities with potential human health benefits, most remarkably anti-inflammatory [11,12,13], anticancer [14] and anti-oxidant activities [15,16].

Interestingly, PDNVs from different plant species, e.g., ginger rhizome [12,15,17], grapefruit [15,16,18], blueberries [13], broccoli [19], lemon [20] and cabbage [11] were shown to have anti-inflammatory activities in vitro and in vivo. The mechanisms underlying the anti-inflammatory effects of PDNVs include the induction of anti-inflammatory cytokines (IL-10, IL-22), suppression of pro-inflammatory cytokines (IL-6, IL-1β and TNF-α), induction of heme oxygenase-1 (HO-1), activation of nuclear factor (erythroid-derived 2)-like 2 (Nrf2) or the inhibition of ERK/NF-κB signaling pathways [15,20]. The molecules in pristine PDNVs with anti-inflammatory activities could be diverse including miRNAs, secondary metabolites, fatty acids, peptides and proteins. Recent work shed light on the presence of highly abundant miRNAs in edible fruit and vegetable PDNVs that could target mammalian genes encoding inflammatory factors [21]. Plant-derived fatty acids, secondary plant metabolites (curcumin, quercetin, capsaicin, etc.), and carotenoids such as lycopene are long known to downregulate inflammatory responses. If these molecules are inside of native PDNVs, their protected delivery to the site of inflammation could provide a way for effective, precise, and safe therapeutic interventions. However, so far only a few studies identified such compounds in PDNVs [14,22]. Peptides and proteins with anti-inflammatory properties have recently been exploited as anti-inflammatory agents for treatment of inflammatory diseases such as rheumatoid arthritis or Alzheimer’s disease [23]. While protein cargo has been described for several plant PDNVs, to the best of our knowledge, there is no study has been conducted to investigate their potential anti-inflammatory activities.

While a considerable amount of work has been performed to engineer mammalian cell-derived EVs to enhance their uptake efficiency or activity [24,25,26], engineering of PDNVs is less common [27,28]. PDNVs are promising candidates for the delivery of therapeutic agents or poorly soluble natural molecules with known bioactivity, as they efficiently cross physiological barriers [29]. Model proteins, Alexa Fluor 647-labeled bovine serum albumin (BSA) and heat shock protein 70 (HSP70) loaded into grapefruit PDNVs were successfully delivered to human peripheral blood mononuclear cells and colon cancer cells [30]. Moreover, model proteins, grapefruit PDNVs have also been used to deliver DNA expression vectors, siRNA, proteins and chemotherapeutic drugs [18,30,31]. Additionally, it was demonstrated that grapefruit-derived nanovectors were stable at 4 °C for more than one month and were still able to carry and deliver the loaded compound, and maintained anti-inflammatory activity [31].

Natural bioactive compounds have been used as therapeutic medications since ancient times. In this context, curcumin (1,7-bis(4-hydroxy-3-methoxyphenyl)-1,6-heptadiene-3,5-dione) is a bioactive molecule with low intrinsic toxicity that is mainly present in turmeric (*Curcuma longa* L.) rhizomes and infrequently in other *Curcuma* species [32]. Due to the highly hydrophobic property, the bioavailability of curcumin is low which still prevents its clinical application [33,34]. Moreover, curcumin is often used as a model compound for the loading of highly hydrophobic compounds into nanodelivery systems [31,33,35]. Two main strategies are described for loading curcumins into EVs or PDNVs: curcumin-encapsulated/loaded and curcumin-primed vesicles [36]. Curcumin-primed EVs were obtained by isolating them from the cells which have been treated with curcumin beforehand [37,38], while curcumin-encapsulated or curcumin-loaded vesicles are prepared by loading curcumin onto the vesicles after they have been isolated [39,40]. The anti-oxidative, anticancer and anti-inflammatory impacts of both formulations have been confirmed by a number of in vitro and in vivo studies. Regarding the anti-inflammatory activity of curcumin-loaded EVs, Sun et al. reported that incorporation of curcumin into EVs enhance curcumin’s solubility, stability, bioavailability and anti-inflammatory activity in vitro and in vivo [33].

Recently we have reported the isolation and characterization of NVs from tomato (*Solanum lycopersicum* L.) fruit [1,2]. Tomatoes are healthy, versatile and an essential fruit of the Mediterranean diet with associated health benefits, including improved antioxidant defense and reduced risk of inflammatory diseases. For the isolation of PDNVs from fruit homogenate, methods based on centrifugation, filtration, precipitation, and chromatography-based separations or their combinations can be used. Currently, most of the studies employ the differential ultracentrifugation (DUC) method to isolate PDNVs. Tomato PDNVs can be isolated from tomato fruit homogenates at a high yield compared to mammalian cells or other plant resources [1,2]. The crude DUC isolates contain numerous co-purifying structures and molecules as well as the NVs. Size exclusion chromatography (SEC) was shown to be useful to remove the small co-purifying molecules including proteins [1]. Gradient density ultracentrifugation (DGUC), on the other hand was efficient in the separation of PDNVs from the viral particle [2]. It should be highlighted that viruses are often co-purifies with plant PDNVs since they have similar physiochemical characteristics. We have shown that tomato PDNVs are associated to colored (yellow) DGUC bands in the ultracentrifuge tube at the lower density bands than viral particles that commonly present in tomato fruit. Moreover, DGUC efficiently separates bulk NVs into different vesicle subpopulations based on their buoyant density. In this paper, we have isolated bulk PDNVs from tomato by GUC. Relying on our previous experience [2] we employed proteomics to confirm that the bulk isolate did not contain viruses. Bulk virus free tomato PDNVs then were separated by DGUC and the main vesicle containing fractions were analyzed regarding anti-inflammatory activity on a macrophage model system. A combined shotgun proteomics and in silico study was performed with the aim to point out candidates that could have an effect on the inflammatory pathways. Moreover, aiming to increase the anti-inflammatory activity, DGUC fractions that were the most abundant in nanovesicles were loaded with curcumin using three different methods, i.e., direct loading, sonication and extrusion. Entrapment efficiency (EE%), drug loading capacity (DL%) and anti-inflammatory activity of curcumin loaded DGUC purified PDNVs were analyzed and compared.

## 2. Materials and Methods

### 2.1. Isolation of Vesicles from Tomato Fruit by Differential Ultracentrifugation

Nanovesicles were isolated from Piccadilly variety tomato fruits (200 g, Vittoria Colonna s.r.l., Sicily, Italy) by differential ultracentrifugation (dUC) according to Bokka et al. [1]. In brief, after washing and removing exocarp, tomatoes were homogenized in a kitchen mixer homogenizer by 3 cycles of 10 s in extraction buffer composed of 100 mM phosphate, 10 mM ethylenediaminetetraacetic acid (EDTA) (J.T. Baker, Deventer, The Netherlands) (pH 8, 170 mL) containing protease inhibitors 1 mg/mL leupeptin (0.085 mL, AppliChem, Darmstadt, Germany), 100 mM phenylmethylsulfonyl fluoride (PMSF, 0.425 mL) and 1 M sodium azide (0.272 mL) AppliChem, Darmstadt, Germany). The isolation of NVs were performed using four low-velocity centrifugation steps at increasing centrifugal force, i.e., 400× *g*, 800× *g*, 2000× *g* and 15,000× *g* each of them for 30 min at 22 °C. Supernatant from the last low-velocity centrifugation step was centrifuged at 100,000× *g* for 2 h at 4 °C using an SW28Ti rotor in Beckman Coulter Optima L-90K ultracentrifuge (Beckman Coulter, Brea, CA, USA). The pellet containing the NVs was re-suspended in a small volume of buffer.

### 2.2. Separation of Nanovesicles into Subpopulations by Density Gradient Ultracentrifugation

NV sample was subjected to density-based fractionation by DGUC on iodixanol gradient media as previously described [2].

Iodixanol density gUC was performed by loading the sample onto 5, 10, 20, and 40% (*v*/*v*) iodixanol (OptiPrep™ (60% (*w*/*v*), Serumwork, Bernburg AG, Germany) solution in a 12.5 mL polypropylene tube and centrifuging at 100,000× *g* for 18 h, at 4 °C using an SW41Ti rotor (Beckman Coulter Optima L-90K Ultracentrifuge). Twelve fractions of 1 mL each were collected from top to bottom. The fractions were washed using extraction buffer by centrifuging the samples at 100,000× *g* for 2 h at 4 °C to remove iodixanol. After the washing step, the pellets were re-suspended in the extraction buffer.

### 2.3. Protein Quantification and SDS-PAGE Analysis

The protein concentration was measured by the Qubit Protein Assay Kit (Thermo Fisher Scientific, Rockford, IL USA). Sodium dodecyl sulfate-polyacrylamide gel electrophoresis (SDS-PAGE) was used for the separation of proteins. Samples (10 µg of proteins) were loaded and electrophoretically separated on a Novex Bolt 4–12% Bis-Tris Plus gel (Invitrogen, Carlsbad, CA, USA), using Bolt MOPS SDS running buffer (Invitrogen, Carlsbad, CA, USA) according to the manufacturer’s instructions. The gel was stained with colloidal Coomassie Brilliant blue G-250 (AppliChem, Darmstadt, Germany) overnight and rinsed with MilliQ water until the background became clear.

### 2.4. Density Determination

Density determination of each fraction was based on the iodixanol concentration measured by Nanodrop 2000 spectrophotometer (Thermo Fisher Scientific Inc., Waltham, MA, USA) at 244 nm wavelength. Measurements of the standards to create the calibration curve and the samples were performed according to the described method [2].

### 2.5. Nanoparticle Tracking Analysis (NTA)

Crude NV samples and each fraction collected after iodixanol DGUC were analyzed by nanoparticle tracking analysis (NTA), as described by Adamo G. et al. [41]. Briefly, NTA measurements were performed using a NanoSight NS300 (Malvern Panalytical, Malvern, UK) equipped with a 488 nm laser, a high sensitivity sCMOS camera and a syringe pump. Tomato NVs were diluted in particle-free water (Water, HPLC grade, Sigma-Aldrich, Saint Louis, MO, USA) and filtered by 20 nm Anotop filters (Whatman, Maidstone, UK) to reach a particle concentration of 1–10 × 10^8^ particles/mL. Following the Malvern’s instructions and our optimization, five videos of 60 s duration were recorded for each sample, using a camera level of 15–16 and a syringe pump speed of 60 A.U., corresponding to 1.5 µL/s. A total of 1500 frames per sample were analyzed by NTA 3.4 Build 3.4.003 software. Instrument-optimized detection threshold, which is the minimum brightness of tracked particles allowing the detection of as many “valid” particles as possible (count number of red crosses should be between 10 and 100 per frame) was used with no more than five rejected tracks per frame (blue crosses).

### 2.6. Lysis of Vesicles and Proteolytic Digestion

Crude and fractionated NVs isolated and purified by dUC and gUC, respectively were subjected to in-solution digestion and LC-ESI-MS/MS analysis. For in-solution digestion proteomics analysis, samples (30 µg of protein measured by Qubit assay) were re-suspended in 0.2% RapiGest detergent (Waters Corporation, Milford, MA, USA) in 50 mM ammonium bicarbonate (AMBIC) and vesicles were lysed applying five cycles of freezing for 5 min in liquid nitrogen and thawing under sonication in an ultrasonic wash bath for 5 min. Proteins were reduced by 5 mM dithiothreitol (DTT, Sigma-Aldrich, Saint Louis, MO, USA) in 50 mM AMBIC and alkylated using 15 mM iodoacetamide (Sigma-Aldrich, Saint Louis, MO, USA) in 50 mM AMBIC. Proteolytic digestion of proteins was performed using trypsin (mass spectrometry-grade from Pierce, Thermo Fisher Sci Rockford, IL, USA) at 100:1 protein to trypsin molar ratio overnight at 37 °C. RapiGest was cleaved by acidifying the samples using trifluoroacetic acid (TFA) to 0.5% final concentration, and the samples were centrifuged at maximum velocity for 10 min. Pellet was removed and the supernatant containing the tryptic peptides were vacuum dried.

### 2.7. LC-ESI-MS/MS

Crude NVs and all the gUC fractions were analyzed by shotgun proteomics as it was described before [2]. Briefly, peptides were purified and enriched using Pierce C18 Spin Columns according to the manufacturer’s description. Eluted samples were dried using SpeedVac and kept at −20 °C. Prior analysis, samples were dissolved in 2% acetonitrile (*v*/*v*), containing 0.1% (*v*/*v*)formic acid (FA). A Waters Acquity I-Class ultra-performance liquid chromatography (UPLC) system equipped with a Waters Acquity CSH Peptide C18 UPLC column (1 mm × 150 mm, 1.7 µm) was employed to separate the peptides. Gradient elution was used with the following parameters: eluent A: 0.1% FA, eluent B: 0.1% FA in acetonitrile; flow rate: 20 µL/min; column temperature: 45 °C; gradient: 1 min: 5% B, 45 min: 35% B, 46 min: 85% B. MS experiments were performed according to the method described in [2].

### 2.8. Bioinformatics

Basic Local Alignment Search Tool (blast) was used to analyze crude NVs and DGUC fractions against non-redundant protein sequence (not redundant v5) in OmicsBox 2.2.4 software package (BioBam Bioinformatics S.L. Valencia, Spain). Input protein sequences in FASTA format were obtained using UniProt ID mapping function against UniProt Ref90 database. Functional annotation of identified proteins was performed using OmicsBox 2.2.4 by selecting (gene ontology) GO terms from the GO pool obtained by the Mapping step and assigning them to the query sequences. GO annotation was carried out by applying an annotation rule on the found ontology terms. InterPro, enzyme codes, KEGG pathways, GO direct acyclic graphs (DAGs) were performed by OmicsBox. Venn diagrams were prepared using the FunRich software (version 3.1.3).

### 2.9. Determination of Lipid Content

To quantify the unsaturated fatty acids in vesicles, the colorimetric reaction of sulphuric acid and phospho-vanillin with lipids (SPV assay) was used in a 96-well plate format. The method was modified from [42]. Briefly, lipid standard solutions (2 μg/μL) were prepared from cholesterol (Sigma-Aldrich, Darmstadt, Germany)/menhaden fish oil (Sigma-Aldrich, Darmstadt, Germany) in chloroform (Romil, Deltek, Pozzuoli, Italy) and different volumes (0, 1, 2, 4, 7 and 14 μL) of standard solutions were pipetted into glass vials. NVs sample containing 10 μg of protein were dried in 1.5 mL Eppendorf tubes using a vacuum concentrator (Savant). A small volume of chloroform was added onto the dried sample, vortexed, and transferred to a glass vial. Chloroform was added onto standard solutions and samples in each vial up to 14 μL final volume. Chloroform was evaporated by incubating the vials at 60 °C in a heater block (Reacti-Therm™ Heating and Stirring Module TS-18822, Thermo Fisher Scientific). A volume of 200 μL of 96% sulphuric acid (Sigma-Aldrich, Darmstadt, Germany) was added to all standards and samples in glass vials. The samples were incubated with open lids at 90 °C for 20 min in a heater block and cooled down to room temperature by placing them for 5–10 min at 4 °C. One hundred and twenty (120) μL 1 mg/mL phospho-vanillin reagent (the solution of vanillin (Sigma-Aldrich, St. Louis, MO, USA) in 17% phosphoric acid (Deltek, Pozzuoli, Italy)) was added into each vial and vortexed. Two hundred and eighty (280) μL of sample was transferred to a 96-well plate (92196, TPP Techno Plastic Products, Trasadingen, Switzerland) and incubated for 1 h at 37 °C to develop the color reaction. The absorbance was read at 520 nm with a VIS plate reader (AMR-100 Microplate Reader) and the amount of lipids was calculated.

### 2.10. Preparation of Curcumin-Loaded Small Unilamellar Vesicles

#### 2.10.1. Cargo Loading by Extrusion

Tomato MVs or DGUC NVs Fraction 8 were vacuum dried (Speed Vac, SC100, Savant Instruments, Inc., Farmindale, NY, USA), and solubilized in 500 µL of chloroform. Curcumin (Sigma Aldrich, St. Louis, MO, USA) was solubilized in methanol at a concentration 5 mg/mL. Different vesicle and curcumin concentrations were used. Solvents were removed by drying the sample under a stream of nitrogen followed by drying in a vacuum desiccator overnight to form a lipid film. Sample was rehydrated using the isolation buffer composed of 100 mM phosphate and 10 mM EDTA, and containing 5% DMSO, vortexed and sonicated and was placed on a low speed rotating wheel at 40 °C overnight. The sample was then centrifuged at 500× *g* for 5 min at 40 °C, the supernatant was collected, filtered through a 0.45 µm pore size filter and extruded through a 0.2 µm pore size polycarbonate membrane by 21 passes with an Avanti Mini-Extruder (Avanti Polar Lipids, Inc., Alabaster, AL, USA). This step was repeated using a 0.1 µm pore size polycarbonate membrane. Tomato MVs sample was washed by size exclusion chromatography after extrusion. The protein concentration was measured before and after extrusion by Qubit assay. The quantity of the incorporated curcumin in vesicles was determined by Nanodrop 2000 UV-Vis spectrophotometer (Thermo Fisher Scientific Inc., Waltham, MA, USA) at 426 nm wavelength. Extruded samples containing 20 µg of proteins were vacuum dried and resuspended in 10 µL of methanol for the determination of curcumin concentration. A standard absorption curve of curcumin in methanol was prepared at seven different concentrations between 0.005 and 0.05 mg/mL. The quantity of incorporated curcumin in the sample was calculated based on the measured absorbance values. The same loading procedure was performed to prepare the blank sample in the absence of curcumin.

#### 2.10.2. Cargo Loading by Sonication

Tomato NVs DGUC Fraction 8 (0.85 mg by protein quantity, 6.7 mg by weight of dried sample) was vacuum dried and 3.4 mg of curcumin (*w*/*w* protein in NVs: curcumin 1:4) was added. Sample was suspended in 200 µL of extraction buffer, and incubated for 1 h at room temperature. Then, it was ultrasonicated in a bath sonicator for 30 min. The sample was centrifuged at 8000× *g* for 5 min and the supernatant containing the loaded NVs and the pellet containing the non-loaded curcumin and NVs aggregates were collected. Protein and curcumin concentrations were determined by Qubit assay and UV-Vis spectrophotometry, respectively.

#### 2.10.3. Passive Cargo Loading

Tomato NVs DGUC Fraction 8 (1.4 mg by protein quantity, 18 mg by dried weight) was vacuum dried, combined with 7 mg of curcumin, solubilized in 100 µL of extraction buffer, and incubated overnight at room temperature. The sample was loaded in a 12.5 mL polypropylene tube containing 8, 30, 45, 60% sucrose cushions and centrifuged at 100,000× *g* for 18 h, at 4 °C using an SW41Ti rotor according to the method described in [2]. Six fractions were collected. Fractions were washed using the isolation buffer and centrifuged again at 100,000× *g* for 2 h at 4 °C. After the washing step, the pellets were re-suspended in the extraction buffer and the protein and curcumin quantities were determined by Qubit assay and UV-Vis spectrophotometry, respectively.

### 2.11. Cell Cultures

All experiments were performed with THP-1 cells, a human leukemia monocytic cell line displaying macrophage-like activity. THP-1 cells were cultured in RPMI 1640 medium (Corning, New York, NY, USA) supplemented with 100 U/mL penicillin, 100 μg/mL streptomycin, 1.2 mM glutamine, and 10% fetal calf serum in an atmosphere of 95% air-5% CO_2_ at 37 °C. THP-1 cells were cultured to a final density of 1 × 10^6^ cells/mL. Cells were pre-incubated in serum free RPMI 1640 medium containing MVs or selected DGUC fractions of NVs from 2 to 10 µg/mL.

### 2.12. MTT Assay and Trypan Blue Staining

To assess THP-1 viability, cells were pre-incubated for 24 h with PDNVs, seeded, and after additional 24 h of growth, methyl thiazolyl tetrazolium (MTT) assay or Trypan blue staining was performed. For MTT assay, medium was removed from each well, cells were incubated with a 500 µg/mL MTT solution for 3 h at 37 °C, 5% CO_2_, and washed. The formazan salts formed were dissolved in 100 μL of DMSO. The absorbance was measured at 570 nm using the Bio-rad model 680 microplate reader. For trypan blue staining, 0.4% trypan blue solution was mixed with cells at 1:1. A LUNA-II™ Automated Cell Counter (Logos biosystem, Anyang-si, Republic of Korea) was used to count the cell numbers. Results reported as survival rate, i.e., the ratio of treated and untreated viable cells (unstained cells) expressed in percentage.

### 2.13. Anti-Inflammatory Activity Assay

THP-1 were cultured to a final density of 1 × 10^6^ cells/mL treatment condition in Roswell Park Memorial Institute Medium (RPMI) cell culture media. Cells were pre-incubated with curcumin (Sigma-Aldrich, Saint Louis, MO, USA), selected pristine DGUC fractions of NVs or curcumin loaded PDNVs fractions. After 4 h of pre-incubation, cells were subjected to an inflammatory stimulus, adding the medium containing 10 and 50 ng/mL of LPS for an additional 4 h. Cells pre-incubated in RPMI media depleted in exosome and stimulated with LPS (without pre-incubation with PDNVs) represented the positive controls. The untreated cells represented the negative controls. Experiments were performed in triplicate.

### 2.14. Inflammatory Cytokine Test

Total RNA from treated cells was extracted according to the manual’s instructions of RNAZOL direct clean up kit (Fisher Molecular Biology). To digest contaminant DNA, RNA samples were treated with RNase-Free DNase I set (VWR International, Milan, Italy). The quantity of total RNA was measured by RNA absorbance measurements using in a microplate reader (SpectraMax QuickDrop Micro-Volume, Molecular Devices, San Jose, CA, USA) at 260 nm wavelength. The PrimeScript™ RT Reagent Kit (Takara Bio Europe, Saint-Germainen-Laye, France) was used to retro-transcribe total RNA (200 ng) according to the manufacturers’ instructions. Complementary DNA products were amplified using a Step One Plus™ RT-PCR System (Applied Biosystems, Monza, MB, Italy). List of the primers used for cDNA amplification: 36B4 as reference gene (Forw: 5′-CGACCTGGAAGTCCAACTAC-3′, Rev: 5′-ATCTGCTGCATCTGCTTG-3′), IL-1β (Forw: 5′-AAAGAAGAAGATGGAAAAGCGATT-3′, Rev: 5′-GGGAACTGGGCAGACTCAAATTC-3′), IL-6 (Forw: 5′-GGTACATCCTCGACGGCATCT-3′, Rev: 5′-GTGCCTCTTTGCTGCTTTCAC-3′). Each reaction was performed in triplicate. The lack of primer dimers was confirmed by melt-curve analysis for each amplification. Relative quantification of expression levels was calculated according to the ΔCq method using 36B4 gene expression as a reference.

### 2.15. Statistics

Results are presented as means ± standard deviation (SD). A one-way ANOVA for repeated measures was used for MTT, Trypan blue and Real-time PCR (IL-1β and IL-6 quantifications) assays, with Dunnett’s post-hoc analysis. The level of significance was *p* ≤ 0.05 (*), *p* ≤ 0.01 (**) and *p* ≤ 0.001 (***).

## 3. Results

### 3.1. Tomato-Derived Vesicles Isolation, Separation Based on Density and Characterization

NVs were isolated from Piccadilly tomatoes by the differential ultracentrifugation (dUC) method. The yield of NVs expressed in protein quantity was 26 ± 11 mg of protein per 1 kg of fruit. The average size of NVs and particle concentration were assessed by NTA and they were 160 nm ± 3 nm and 5.8 × 10^10^ particles/µg of protein content, respectively. NVs were separated into 12 density fractions by DGUC using sucrose or iodixanol gradient media (Figure 1a) and the DGUC fractions were characterized by particle size distribution, concentration, morphology, density, protein and lipid quantities, and SDS-PAGE protein profiles (Figure 1 and Figure 2). Three orange color bands (B) were visible after DGUC: band 1 (B1) mainly in Fraction 5, band 2 (B2) in Fraction 8 and B3 in Fraction 9 (Figure 1a). Cryogenic transmission electron microscopy (cryo- TEM) confirmed the presence of small membrane limited structures heterogeneous in size and shape present in the visible bands (Figure 1b). The densities of visible Fraction 5, Fraction 8 and Fraction 9 were 1.064 ± 0.007 g/mL, 1.103 ± 0.006 g/mL and 1.122 ± 0.012 g/mL, respectively (Figure 1c), slightly lower than the reported density of mammalian EVs (1.13–1.19 g/mL) [43]. The particle size distributions show that the lower-density Fractions 1–8 contained slightly bigger nanoparticles (180 ± 4 nm) than the higher-density Fractions, 9–12 (146 ± 3 nm), the smallest particles were observed in Fraction 11, and the largest particles in Fraction 1 (Figure 1d).

The highest protein quantity was measured in Fraction 8 which corresponded to 20.8% of the total protein amount loaded (Figure 2a) while in Fraction 1, 11 and 12 we detected only low amounts of proteins. The highest number of particles was observed in Fraction 8 (9.8 × 10^12^) (Figure 2b) and considerably less in the other two visible bands, i.e., Fraction 5 (3.84 × 10^11^) and Fraction 9 (1.92 × 10^12^). Figure 2d shows the vesicles numbers normalized for protein amount (1 µg). In this respect, amongst the tree visible bands, fraction 5 exhibited the highest number of particles to protein ratio.

Protein profiles in the crude NVs and the fractions were followed by SDS-PAGE analysis (Figure 2c). Crude NVs show a characteristic, reproducible and very complex protein pattern, which became more resolved after DGUC fractionations (Figure 2c).

The unsaturated lipid quantity in crude NVs, and in the three visible bands (Fraction 5, 8 and 9) were determined by the sulpho-phospho-vanillin (SPV) assay. The amount of unsaturated lipids in the three visible bands were proportional to their measured protein quantity (Figure 2a). For example, Fraction 8 contained 398 ± 58 µg lipids and 420 ± 156 µg of proteins where the other two visible bands contained less lipids and proteins.

### 3.2. Proteomic Characterization of Tomato Fruit-Derived NVs and DGUC Fractions

In-solution digestion and LC–ESI–MS/MS-based shotgun proteomic analysis was performed to define the protein biocargo of crude NVs and the 12 DGUC fractions. Figure 3a shows the number of identified proteins and Appendix A reports the details on the identification and functional annotation relative to each protein in the DGUC fractions and the DUC isolate. In the first and the last two DGUC fractions (Fraction 1, Fraction 11 and 12) which were associated to low protein and lipid contents (Figure 2a) as well as low particle numbers (Figure 2b) there was only one protein identified (Appendix A). The crude NV sample analyzed before fractionation resulted in 220 identified proteins (Appendix A). To aid the functional analysis, blast search and GO term annotation were performed and the resulted protein descriptions are reported in Appendix A together with the GO terms, enzyme codes and enzyme’s names. Proteomic analysis of the three visible DGUC bands yielded 258 (Fraction 5), 190 (in Fraction 8) and 175 (in Fraction 9) proteins. The Venn diagram (Figure 3b) shows the 82 overlapping (common proteins) proteins over these fractions. The high numbers of unique proteins, 138 in Fraction 5, 46 in Fraction 8, and 35 in Fraction 9 indicate that the different density fractions have different protein contents. In particular, we have found that the low-density Fraction 5 (band 1) was considerably different in protein content from the two higher density visible bands (Figure 3b). Linoleate 9S-lipoxygenase B (Q42873_SOLLC, 9-LOX) was the highest-ranking protein identified in both crude NVs and the visible DGUC fractions. Lipoxygenases (LOXs) are naturally occurring widely distributed enzymes in plants and animals. There are at least five LOXs in tomato plant and one has been purified and characterized from the supernatant fruit homogenate [44]. In our previous work, we have already identified LOX in tomato PDNVs [1,2] as highly expressed protein. In addition to 9-LOX, here we have identified other less-expressed LOXs too (A0A3Q7EN25_SOLLC, P38416|LOXB_SOLLC, Q42873_SOLLC, Q9FT17_SOLLC, A0A3Q7ENA4_SOLLC, A0A3Q7HN01_SOLLC). The lipoxygenase isoforms are distinguished by differences in substrate and product specificity. The LOX family proteins catalyze the stereo and region-specific addition of oxygen molecule to the 1,4-cis,cis-pentadiene segment of substrate lipids, i.e., linolenic acid, α-linolenic acid, and linoleic acid. Plant lipoxygenases are known to be involved in diverse aspects of plant physiology including growth and development, resistance to pathogen attracts, and response to wounding. Lipids are key components in the biogenesis and bioactivities of membrane vesicles, the association of LOX to the tomato NVs is interesting. Oxylipins of linoleic acid can be both anti-inflammatory and pro-inflammatory, often associated with atherosclerosis, non-alcoholic fatty liver disease, and Alzheimer’s disease [45]. Further investigations must be carried out to unravel the contribution of NVs associated LOX and their role in plant physiology. In addition, mammalian exosomes are also known to contain LOX key enzymes that acts on the leukotriene pathway [46] and some of them are involved in inflammatory processes.

In addition to LOXs, V-ATPases (Q84XW6_SOLLC, Q84XV7_SOLLC, A0A3Q7FE06_SOLLC), carboxylesterase (K7SGP9_SOLLC, A0A0C5CHE6_SOLLC, G8D593_SOLLC), heat shock cognate proteins (HSP80_SOLLC, Q6UJX4_SOLLC, A0A3Q7FX57_SOLLC, A0A3Q7IYI9_SOLLC), patellin-3-like (A0A3Q7GYN8_SOLLC, A0A3Q7IXE6_SOLLC) and vicilin-like proteins (B0JEU3_SOLLC, A0A3Q7I6G7_SOLLC, A0A3Q7J1U2_SOLLC) were amongst the top-ranking proteins in NVs and in DGUC fractions 2–10. Most of these proteins are commonly associated to mammalian EVs and other PDNVs.

Next, we were interested to find proteins within the tomato PDNVs dataset (Appendix A) that putatively effect the mammalian cells’ response to inflammatory processes. Inflammation is the response of innate immune system to injury, pathogen attack, damaged cells or toxic compounds, etc., characterized by the release of pro-inflammatory molecules and anti-inflammatory cytokines. We supposed that proteins in PDNVs could modulate inflammation in various ways. One way could be through protein regulators analogous to human regulatory proteins participating in the inflammatory process. Therefore, we have extracted and analyzed key proteins relative to NVs, DGUC Fraction 5, 8 and 9 that are involved in “Regulation of cellular processes” GO:0050794 term (Appendix A). There were 21 proteins in NVs, 30 proteins in Fraction 5, 24 proteins in Fraction 8 and 19 proteins in Fraction 9 associated with GO:0050794 term (Appendix A). Several of these proteins were commonly expressed in the different DGUC fractions. Figure 4 shows how the child GO terms are structured as acyclic directed graphs in the three fractions analyzed. The highest number of sequences associated to GO:0050794 was obtained in Fraction 5, while the highest node score was obtained in Fraction 8 (Figure 4).

Most of the proteins listed in Table 1 participate in signal transduction processes. These are proteins that can trigger a change in the activity or the state of a cell. In the tomato PDNV (Fraction 8) these were eight members of 14-3-3 protein family (P93209, P93207, P93214, P93206, Q41418, A0A3Q7EAX2, P93212 and A0A3Q7JBH3), a mitogen activated protein kinase (A0A3Q7JBH3), a senescence associated protein (A0A3Q7HZY2), a AAA domain containing protein (A0A3Q7F894) and three luminal binding proteins (BiPs, A0A3Q7GTI1, Q9FSY7 and A0A3Q7GTI1). The 14-3-3 family are highly conserved multifaceted proteins that regulate the innate and adaptive immune responses by modulating gene expression, signal transduction, inflammatory mediator level and immune cell activity [47]. The luminal binding proteins (BiPs) is one of many HSP-70 proteins very well conserved over all organismal kingdoms. This important chaperon has known anti-inflammatory and immunomodulatory properties and therefore it was proposed as biologic for rheumatoid arthritis [48].

### 3.3. Cytotoxicity of Tomato-Derived NVs

Methyl thiazolyl tetrazolium (MTT) and Trypan blue staining assays are widely used methods for determining cell viability upon treatment with PDNVs. The MTT assay showed decreasing MTT values by increasing vesicle concentrations for all three fractions studied indication a slight toxic effect, especially at 10 µg (Figure 5a). As shown in Figure 5b, after 48 h of THP-1 cell incubation with tomato fruit NVs, in more detail, Fraction 5, Fraction 8, or Fraction 9 at a concentration ranging from 2 to 10 µg/mL, trypan blue assay did not reveal alterations of cell viability between control and treatment conditions. Nonetheless, it is possible to appreciate a tendency of 10 µg PDNV conditions to slightly decrease the percentage of viable cells, even if not at statistically significant levels.

### 3.4. Anti-Inflammatory Activity of Native Tomato NVs on THP-1 Cell Line

First, we aimed at evaluating the in vitro anti-inflammatory activity of pristine tomato-derived NVs. For this, the main vesicle populations, i.e., the three visible bands corresponding to DGUC Fractions 5, 8 and 9 have been investigated. In our experiments (Figure 6), the cells were pre-incubated with NVs fractions for 4 h. Then, inflammation was induced by 10 or 50 ng/mL of lipopolysaccharide (LPS) for 4 h and the mRNA expression levels of pro-inflammatory cytokine IL-1β were measured by Real-Time quantitative PCR. Cells were treated with either 2 or 5 µg/mL vesicles (measured as protein quantity) corresponding to 1.01 × 10^11^ and 2.53 × 10^11^ number of particles for Fraction 5, 5.27 × 10^10^ and 1.32 × 10^11^ number of particles for Fraction 8, and 4.46 × 10^10^ and 1.12 × 10^11^ number of particles for Fraction 9, respectively, measured by NTA.

We observed a dose-independent down-regulation of the expression of pro-inflammatory cytokine IL-1β mRNA in DGUC Fractions 5 and 8 at 10 and 50 ng/mL of LPS stimulation (Figure 6b,c). In contrast, DGUC Fraction 9 failed to down-regulate IL-1β mRNA expression (Figure 6d). Comparing these fractions, Fraction 8 displayed the highest anti-inflammatory effect during the LPS-induced inflammation process at both concentrations tested (Figure 6c).

### 3.5. Curcumin Loading into Tomato Vesicles

To increase the baseline anti-inflammatory activity of pristine tomato NVs, we tempted to load curcumin, a lipophilic natural compound with known anti-inflammatory activity into tomato vesicles. Three different loading methods were applied: direct incubation, sonication and extrusion, according to Figure 7 and Table 2. DGUC Fraction 8 was chosen for the curcumin loading experiments because this was the most abundant fraction in terms of nanoparticles, protein and lipid concentrations (5.27 × 10^10^ particles/mL, 2.4 µg of proteins/µL, 2.57 µg of lipids/µL) and because it showed the highest baseline anti-inflammatory activity within the pristine NV subpopulations (Figure 6c). We also tried to load MVs (only by extrusion), since this abundant fraction was rich in lipids and showed similar characteristics to NVs. Different vesicles (µg in proteins) to curcumin (µg) mass ratios were used from 5:1 to 1:10 (Table 2).

The different loading methods were compared in terms of entrapment efficiency (EE%), drug loading capacity (DL%), and anti-inflammatory activity. EE% and DL are defined by the following formulas [49]:(1)$$\mathrm{EE}\%=\frac{A_{e}}{A_{i}}\times 100\%\mathrm{DL}\%=\frac{A_{e}}{A_{c}}\times 100\%\;\;\;\;\;\mathrm{DL}\%=\frac{A_{e}}{A_{c}}\times 100\%$$
where EE%—entrapment efficiency; DL—drug loading capacity; A_e_—the amount of the entrapped curcumin; A_i_—the amount of the initial curcumin; Ac—the total amount of the NVs carrier (by protein quantity).

Direct loading of vesicles with curcumin yielded the highest entrapment efficiency and loading capacity values compared to the other two methods (Table 2). In this method, we used DGUC on sucrose gradient to separate the loaded NVs from the free curcumin. One yellow colored band between the 30 and 45% sucrose cushions was observed and collected. Extrusion in our hands led to the lowest loading efficiency and frequently caused sample lost too. It should be noted that the ratio of NVs and curcumin was different in the different loading experiments that could also influence our results.

### 3.6. Anti-Inflammatory Activity of Curcumin Loaded Tomato NVs on THP-1 Cell Line

Anti-inflammatory effect of curcumin loaded tomato vesicles was studied in the same way as the pristine PDNVs using 10 ng/mL LPS stimulation and analyzing mRNA expressions of pro-inflammatory cytokines IL-1β and IL-6. The amounts of curcumin that were loaded into tomato vesicles with three different methods are reported in Table 2. As a control, THP-1 cells were treated with free curcumin at two different concentrations 0.048 and 7 µg/mL. The lower concentration represents the minimum curcumin level that we were able to load into tomato PDNVs (Table 2) and the higher one corresponds to the quantity of curcumin that is frequently used in in vitro experiment. IL-1β and IL-6 levels showed a very similar trend (Figure 8b,c). High (7 µg/mL) curcumin concentration showed the highest anti-inflammatory activity by almost completely inhibiting the LPS-stimulated expression of IL-1β and IL-6. The lower curcumin concentration (0.048 µg/mL) to a lesser extent but still efficiently decreased the LPS-induced expression of IL-1β and IL-6 (Figure 8b,c).

Curcumin loaded NVs showed anti-inflammatory effect by reducing both IL-1β (Figure 8b) and IL-6 (Figure 8c) mRNA levels. Three different loading methods showed a similar level of the reduction of IL-1β but sample obtained by sonication resulted in a higher reduction of IL-6 mRNA expression.

## 4. Discussion

In this work, by using our previously published dUC procedure we have isolated bulk PDNVs from tomato fruit homogenate for bioassays and loading experiments [1,2] (Figure 1). Most published studies only use bulk PDNVs isolates for downstream loading, cargo analysis and bioactivity assessment. Bulk preparations, however, contain different vesicle populations that are characterized by different biocargo composition, biological activity or can be contaminated by viral particles. Separating the different vesicle populations is difficult and current methods are limited. For example, SEC is very useful to remove low molecular weight co-purifying components but unsuccessful in the removal of viral or lipid particles that have similar size as PDNVs. Density based separation has also its drawback when it is aimed at the separation of particles of similar density but different molecular content or cellular origin. Affinity-based separation can overcome the specificity problem but requires affinity reagents based on molecular markers, which, such as in the case of PDNVs, are not always available for the system under study. In our previous work, we observed that the high-density fractions of tomato PDNVs are often co-purify with viruses [2], which makes their safety use in drug delivery questionable. Here, we have used the density-based DGUC separation using iodixanol gradient media and a non-linear gradient method with a long (18 h) separation time to obtain subpopulations from bulk tomato PDNVs. This method provided higher separation than the previously published sucrose gradient [1] and resulted in three well-separated visible bands (Fraction 5, 8 and 9) observed at somewhat lower densities than the density reported for mammalian EVs. Based on NTA, protein concentration measurements and PSV assay, Fraction 8 was the most abundant in particles. Moreover, we found that these fractions contain unsaturated lipids at protein to lipid ratio about 0.9.

In recent years, studies have demonstrated that nanometer-sized membrane-enclosed particles isolated from several edible plant sources have anti-inflammatory activity [11,12,13], and their cellular uptake, their potential in tissue repair and regeneration and in drug delivery have been studied in vitro and in vivo. In this study, we performed an in vitro study to elucidate the cytotoxic effect and the anti-inflammatory potential of tomato PDNVs for the first time. With respect to previous studies that used bulk vesicle isolates, here we analyzed PDNV subpopulations. We observed that the three tomato-derived PDNVs fractions suppress LPS-induced inflammation in THP-1 cells to a different extent (Figure 5) when treated with a non-toxic level of vesicles. The molecular component with anti-inflammatory activities within the pristine vesicles can be very diverse in chemical nature. Here, we studied the protein content of the vesicles populations and tried to dissect the proteins that could have an effect on inflammatory pathways. While the physiochemical and morphological features of the three subpopulation were similar (Figure 2), proteomics showed a considerable difference between their protein contents (Figure 3). Functional analysis of the proteins identified highlighted several proteins with GO:0050794 term that participate in the regulation of cellular processes. These include a high number of 14-3-3 family proteins, GTP-binging proteins and BiP luminal binding proteins with important regulatory function in pro- and anti-inflammatory processes. Plant secondary metabolites biocargo can also influence the bioactivity of the PDNVs. Regarding tomato, it was reported that it contains a large number of anti-inflammatory compounds and their derivatives. Previously, we showed that phosphatidic acid, a dietary fatty acid known to control the incidence and severity of inflammation, was the most abundant phospholipid identified in tomato PDNVs [1]. Other compounds includes oxylipins products of LOX (highly expressed protein in our preparations, Appendix A), coumarin [50], and terpenes such as lycopene [51] that could also count for the anti-inflammatory effect of tomato-derived PDNVs, but here were not studied.

DGUC Fraction 8 was chosen for the curcumin loading experiments because it exhibited the highest baseline anti-inflammatory activity (Figure 6c). Moreover, this fraction was useful because it contained more nanoparticles than the other two fractions. Here, we used three methods that are frequently used for EV loading, direct loading, loading by sonication and extrusion and compared them based on EE%, DL% and anti-inflammatory effect. In our formulations, we tried four different vesicle to curcumin ratios: 1:2, 1:4, 1:5 and 1:10. The highest EE% (0.22) and DL% (3.60) were obtained at 1:5 ratio (Table 2). EE% and DL% for all different forms of curcumin loaded preparation were lower than expected. We think that it is because of the very low solubility of the curcumin in aqueous solution used for the preparation and may be improved by solvent exchange or the addition of detergent. The anti-inflammatory effect measured as the fold change of pro-inflammatory cytokines IL-1β was similar for the curcumin PDNVs loaded with the three different methods (Figure 7), but the IL-6 mRNA level was more reduced when cells were treated with vesicles loaded by curcumin with sonication (Figure 8). The higher efficiency could due to the vesicle size, or integrity. The current study represents a preliminary investigation for the successful encapsulation of curcumin in tomato-derived PDNVs and further studies are necessary to better understand the underlying mechanisms.

## Figures and Tables

**Figure 1 pharmaceutics-15-00333-f001:**
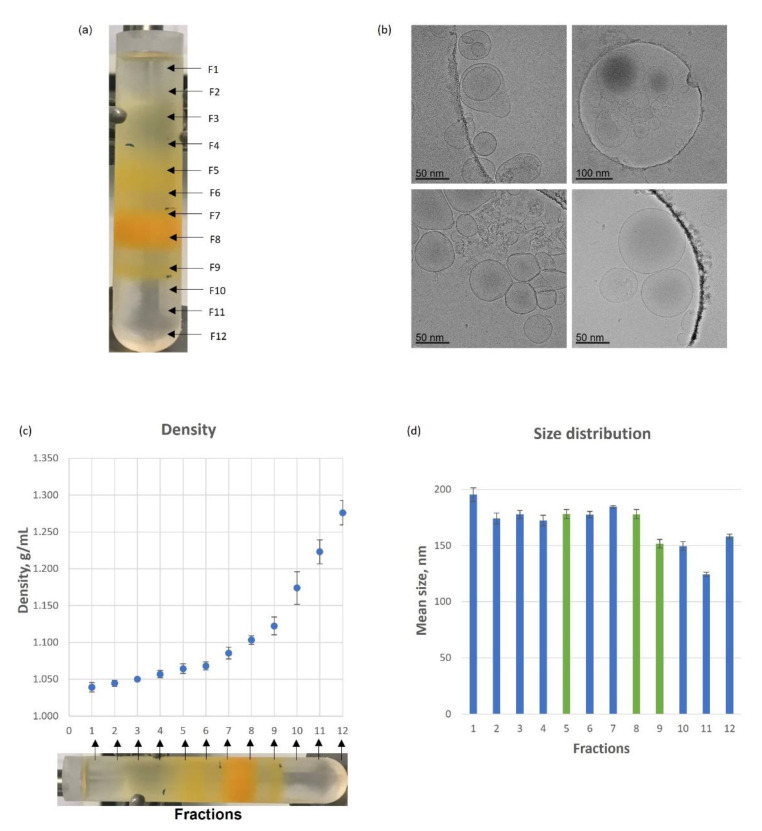
Separation of tomato-derived nanovesicles (NVs) into subpopulations using iodixanol density gradient ultracentrifugation (DGUC). (**a**) NVs (5 mg in protein content) were loaded and 12 fractions, F1–F12 were collected, three colored bands at Fraction 5, 8 and 9 were observed; (**b**) representative image of cryogenic transmission electron microscopy (cryo-TEM) images of Fractions 5 and 8; (**c**) calculated densities of each fraction, *n* = 6); (**d**) Mean size of the particles in each fraction.

**Figure 2 pharmaceutics-15-00333-f002:**
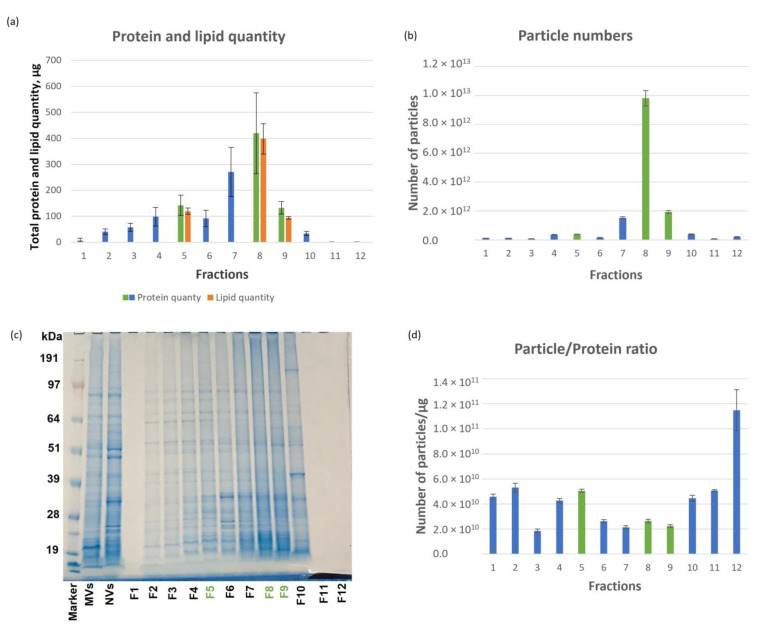
Physical and molecular characteristics of the tomato NVs subpopulations. (**a**) The distribution of protein quantity over the iodixanol density gradient ultracentrifugation (DGUC) fractions (normalized for 1 mg of loaded NVs sample, *n* = 3); (**b**) number of particles in each fraction; (**c**) sodium dodecyl sulfate–polyacrylamide gel electrophoresis protein profiles of tomato NVs and fractions separated by iodixanol DGUC (the loaded protein quantity for each fraction was 10 µg, except Fraction 1, 11 and 12, for these fractions the protein amount was around 1 µg); (**d**) the particle to protein ratio in each fraction.

**Figure 3 pharmaceutics-15-00333-f003:**
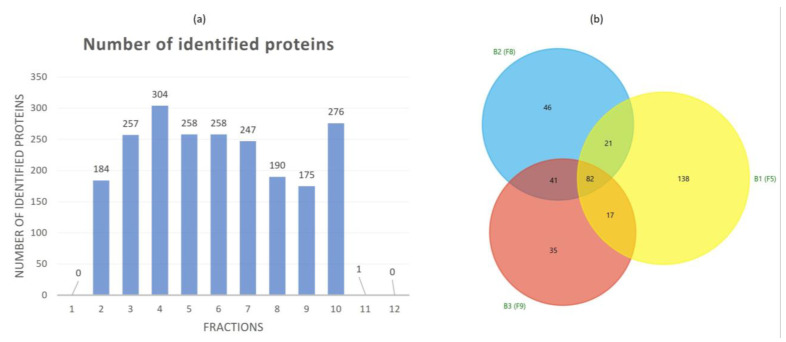
(**a**) Number of total identified proteins in each fraction separated by iodixanol density gradient ultracentrifugation (DGUC); (**b**) Venn diagram of the proteins identified in the proteomics study showing a similarity between the identified proteins in different DGUC fractions.

**Figure 4 pharmaceutics-15-00333-f004:**
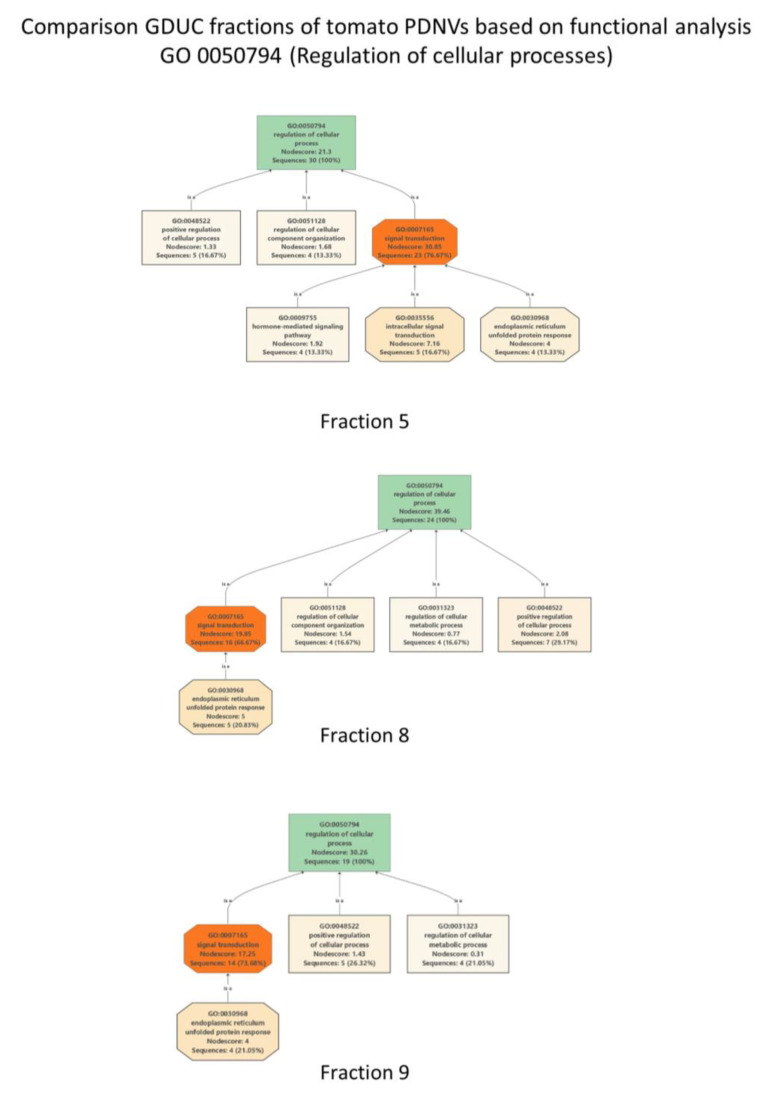
Acyclic directed graphs of GO:0050794 term in density gradient ultracentrifugation separated visible fractions (Fraction 5, Fraction 8 and Fraction 9) of tomato nanovesicles. Graph was filtered by four sequences. The node score is the sum of sequences directly or indirectly associated to a GO term weighted by the distance of the term to the term of “direct annotation”, i.e., the GO term the sequence is originally annotated to.

**Figure 5 pharmaceutics-15-00333-f005:**
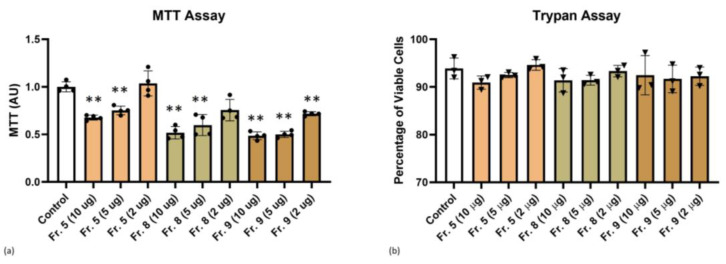
Determination of cell viability by (**a**) MTT and (**b**) Trypan blue assays. The THP-1 cells were treated with different concentrations of tomato-derived PDNVs (2, 5, and 10 µg/mL) separated into three visible fractions, namely Fraction 5 (Fr. 5), Fraction 8 (Fr. 8) and Fraction 9 (Fr. 9) by density gradient ultracentrifugation (DGUC). The treatment was performed for 48 h while control was untreated. ANOVA test followed by Dunnett’s post-hoc analysis. The level of significance was *p* ≤ 0.05 or *p* ≤ 0.01 (**) versus control condition.

**Figure 6 pharmaceutics-15-00333-f006:**
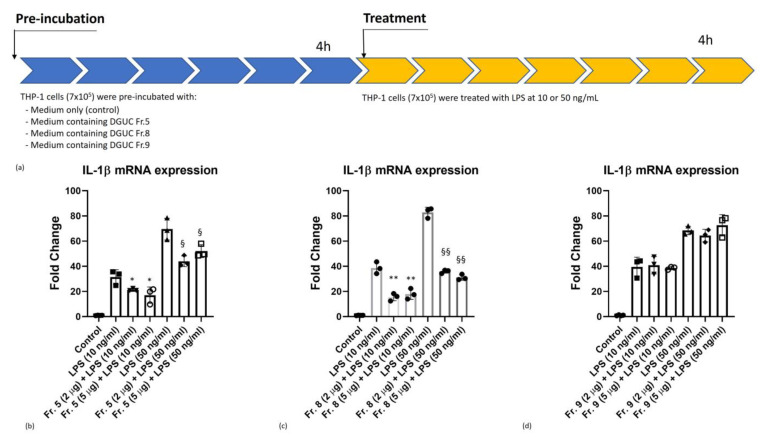
The anti-inflammatory activity of pristine tomato-derived nanovesicles (NVs) (**a**) pre-incubation and treatment conditions, (**b**) density gradient ultracentrifugation (DGUC) Fraction 5; (**c**) DGUC Fraction 8; and (**d**) DGUC Fraction 9 compared with controls. ANOVA test followed by Dunnett’s post hoc analysis. The level of significance was *p* ≤ 0.05 (*), *p* ≤ 0.01 (**) versus LPS (10 ng/mL) condition, and *p* ≤ 0.05 (§), *p* ≤ 0.01 (§§) versus LPS (50 ng/mL) condition.

**Figure 7 pharmaceutics-15-00333-f007:**
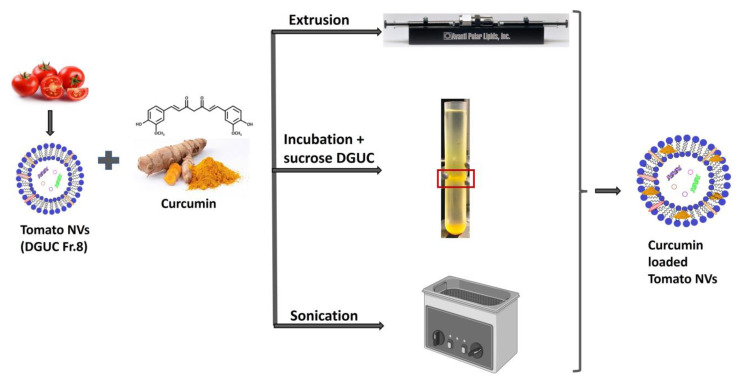
Schematic representation of the process to prepare curcumin loaded tomato nanovesicles by three methods: extrusion, direct loading by incubation followed by sucrose density gradient ultracentrifugation, and sonication.

**Figure 8 pharmaceutics-15-00333-f008:**
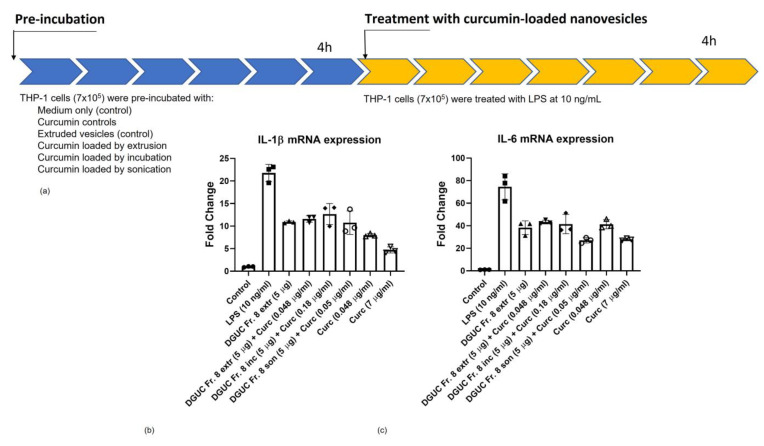
Anti-inflammatory effect of curcumin loaded tomato-derived vesicles. (**a**) Pre-incubation and treatment conditions. Three different methods were used for loading according to Figure 6 and Table 2. As a control, curcumin was used at two different concentrations. Density gradient Fraction 8 was extruded (DGUC Fr. 8 extr., control) or loaded by direct incubation followed by DGUC purification (DGUC Fr. 8 Inc.), sonication (DGUC Fr. 8 son.) and anti-inflammatory effect was evaluated on THP-1 cell line measuring the levels of pro-inflammatory cytokines (**b**) IL-1β; (**c**) IL-6. All the tested conditions were significantly down-regulated compared with LPS (10 ng/mL) condition (ANOVA test followed by Dunnett’s post-hoc analysis; *p* ≤ 0.001).

**Table 1 pharmaceutics-15-00333-t001:** Proteins identified with “Regulation of cellular processes” gene ontology (GO) term (GO:0050794) in Fraction 8 (visible Band 2) of the density gradient ultracentrifugation separated tomato fruit-derived nanovesicles (data extracted from Appendix A).

Cluster ID UniRef	Cluster Name	Ranking in Dataset	mW (Da)	PLGS Score	Coverage (%)	Precursor RMS Mass Error (ppm)	Products	Products RMS Mass Error (ppm)
A0A1U8H482	Alcohol dehydrogenase 1	6	41,275	23,279	50.3	0.9	534	24.4
A0A3Q7GTI1	Luminal-binding protein 5	85	60,108	2618	18.1	3.3	196	36.6
A0A1S4AVH1	GTP-binding protein SAR1A	94	22,021	7487	48.2	4.3	167	29.5
Q9FSY7	Endoplasmic reticulum chaperone BiP	28	73,444	17,203	29.0	5.7	325	33.7
P93209	14-3-3 protein 3	97	30,420	6509	17.9	6.8	125	29.8
Q03685	Luminal-binding protein 5	43	73,859	16,665	23.8	4.9	319	35.9
Q7Y240	Glutaredoxin-dependent peroxiredoxin	161	17,425	4178	22.2	3.3	88	34.1
P93207	14-3-3 protein 10	101	33,680	4477	16.3	1.6	161	34.6
P93214	14-3-3 protein 9	135	29,413	4033	13.8	1.8	158	37.3
P49118	Luminal-binding protein	42	73,189	17,054	22.5	4.9	311	33.4
P93206	14-3-3 protein 1	100	28,183	4550	19.3	3.8	152	34.1
P25858	Glyceraldehyde-3-phosphate dehydrogenase GAPC1, cytosolic	96	36,650	6664	22.8	1.9	189	33.4
Q41418	14-3-3-like protein	63	29,320	8262	38.5	4.5	248	32.3
A0A3Q7EAX2	14-3-3 domain-containing protein	153	28,176	2311	6.4	1.6	103	35.5
P93212	14-3-3 protein 7	134	28,796	4212	17.5	2.0	159	36.5
A0A3Q7ETU0	AAA domain-containing protein	184	48,803	2396	6.7	31.2	137	39.2
A0A3Q7EH12	GTP-binding protein SAR1A	110	21,910	5624	40.4	1.6	135	31.9
A0A3Q7F894	AAA domain-containing protein	162	51,696	4109	6.1	5.6	165	39.7
A0A3Q7HZY2	Senescence-associated protein	149	30,101	3726	19.2	0.8	90	29.4
A0A3Q7GX91	mitogen-activated protein kinase	48	142,212	6454	12.0	5.6	405	36.6
A0A3Q7JBH3	14-3-3 domain-containing protein	70	45,120	7041	16.7	17.3	160	29.0
W1P062	Ras-related protein RABH1b	167	23,083	3665	9.1	6.3	136	39.4
A0A1S3YWY0	Mediator of RNA polymerase II transcription subunit 37a-like	155	74,724	13,707	2.5	18.5	92	35.2
M1CBH0	Alcohol dehydrogenase 1	139	41,124	3025	4.5	6.4	73	21.7

**Table 2 pharmaceutics-15-00333-t002:** Loading of curcumin into tomato vesicles using three methods: extrusion, direct loading and loading by sonication. Entrapment efficiency (EE%) and drug loading capacity (DL%) were determined.

Sample	Loading Method	Ratio PDNVs (µg of Protein:µg of Curcumin)	Quantity of Loaded Curcumin (µg)	Quantity of Loaded PDNVs (by Protein Content µg)	EE%	DL%	Curcumin Quantity Used for In Vitro Assay (µg)
Blank (MVs)	Extrusion		0	77	-	-	-
Blank (DGUC Fr. 8)	Extrusion		0	924	-	-	-
MVs	Extrusion	1:2	0.78	94	0.08	0.82	0.08
DGUC Fr. 8	Extrusion	1:10	5.36	540	0.03	0.98	0.05
DGUC Fr. 8	Incubation	1:5	15.63	418	0.22	3.60	0.18
DGUC Fr. 8	Sonication	1:4	2.78	254	0.10	1.10	0.05

## Data Availability

Not applicable.

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
