# Peer review of "Protein Biocargo and Anti-Inflammatory Effect of Tomato Fruit-Derived Nanovesicles Separated by Density Gradient Ultracentrifugation and Loaded with Curcumin"

_pharmaceutics, 2023, doi:10.3390/pharmaceutics15020333_

Round 1

Reviewer 1 Report

Mammadova et al. investigated plant-derived nanovesicles (PDNVs) from tomatoes to be used as a potential carrier for drug delivery and curcumin was used as a model drug loaded into PDNVs. 12 DGUC fractions of tomato PDNVs were evaluated for their protein profile and anti-inflammatory activity, in which the best fraction was selected for encapsulating the curcumin. The result is significant and the manuscript is well-written. It can be accepted in its current form.

I only have minor revisions:

1/ Line 153-155: please write the concentration and volume properly. For example: 10 mM PBS (pH 7.4, 0.5 mL).

2/ Line 185: please revise “according to [25]” into “according to the described method [25]”.

Author Response

Dear Reviewer 1,

Thank you for your  the revision of our manuscript “Protein biocargo and anti-inflammatory effect of tomato fruit-derived nanovesicles separated by density gradient ultra-centrifugation and loaded with curcumin” by Mammadova et al.. We are glad that you have appreciated our work. We thank you for your valuable comments. According to the comments, we have improved the text as suggested by you. Please find the revised version of our manuscript and our  answer to your two comments:

1/ Line 153-155: please write the concentration and volume properly. For example: 10 mM PBS (pH 7.4, 0.5 mL).

A: We have corrected the buffer concentration, added the volume when it was missing and rephrased the sentence.
2/ Line 185: please revise “according to [25]” into “according to the described method [25]”.

A: We have corrected this.

Thank you again.

Sincerely yours.

Gabriella Pocsfalvi

Reviewer 2 Report

The authors built on a previous study that they conducted and provided further detailed information about the characterization of isolated nanovesicles from tomato homogenate. They loaded the selected fraction with Curcumin and investigated at the in vitro level the anti-inflammatory potential of the nanoformulations.

-        The title should replace (loaded by) with (loaded with).

-        Abstract should include information about curcumin loading and respective results

-        Introduction should be shortened, and more focus should be given on previous use of tomato homogenate rather than ginger as it is the current case

-        Introduction should highlight the limitations of other separation methods and clearly identifies the research gap

-        Multiple references to the same content should be kept to the minimum. For instance, the first line in the introduction has 27 references (1-27)!.

-        In NTA, more details should be given about the (optimum detection threshold)

-        Lysis of vesicles and proteolytic digestion is missing the reference used to develop this method using those conditions of lysis

-        Add relevant references to the section of curcumin loading techniques

-        The experiment of curcumin release from the nanovesicles is missing and should be included.

-        Why in the MTT and trypan blue assays, the groups of NVs, curcumin and curcumin loaded  PDNVs fractions not included to test for their potential in situ toxicity?

-        Figure 4 is very low in resolution and should be enhanced

-        Table 2 needs adjustment since the texting of the sample and loading methods are in the same level

-        Fig 8 depicted different results for LPS (10 ng/ml) and fr 8 (5 µg) than those presented in Fig 6 b for IL-1β although they are the same experimental settings. why is that?

-        Remove the reference from line 663 since the paragraph is about this study

-        Why is the sonication method of curcumin associated with the highest anti-inflammatory response although it does not have the highest EE% or DL%?

-        Assessing the anti-inflammatory response at the mRNA level only is not sufficient to conclude an anti-inflammatory response. Assays at the protein level should be conducted to support this claim.

-        The discussion closely resembles the introduction. It should be rewritten to provide justification to the observed results and comparison to previous closely relevant ones.

-        EE% and DL% for all different forms of Curcumin loaded preparation are too low. In the discussion, possible justification and methods for improvement should be presented. 

Author Response

Dear Reviewer 2, 

Thank you for your critical but very constructive review report. We have revised the manuscript according to your suggestions. In particular we have re-elaborated the introduction and the discussion sections, and we believe that it is now much more focused. Here is out point-by point answers to your points.

our point-by-point answers are as follow

-        The title should replace (loaded by) with (loaded with).

A: We have corrected this error.

-        Abstract should include information about curcumin loading and respective results

A: We have added to the abstract the following information about curcumin loading: The most abundant middle-density DGUC fraction was loaded with curcumin using direct loading, sonication and extrusion methods and anti-inflammatory activity was compared. Highest entrapment efficiency and drug loading capacity was obtained by direct loading. Curcumin loaded by sonication increased the basal anti-inflammatory activity of tomato PDNVs.

-        Introduction should be shortened, and more focus should be given on previous use of tomato homogenate rather than ginger as it is the current case

A: We have shortened and fully revised the introduction based on your suggestion. The revised introduction gives more details on the previous works on tomato homogenate-derived vesicles, delineate the research gaps, the research aims and its novelty respect our previous tomato vesicles related studies.

-        Introduction should highlight the limitations of other separation methods and clearly identifies the research gap

A: See above

-        Multiple references to the same content should be kept to the minimum. For instance, the first line in the introduction has 27 references (1-27)!.

A: We tried to keep the multiple references to the minimum. In the revised paper, the number of the references  was considerably reduced.

-        In NTA, more details should be given about the (optimum detection threshold)

A: Paragraph 2.5 was rewritten and description on optimal detection method has been added to the new version.

-        Lysis of vesicles and proteolytic digestion is missing the reference used to develop this method using those conditions of lysis

A: Vesicle lysis and proteolytic digestion was performed according to our laboratory protocol. We do not have references to add. This paragraph was checked to see if all the described methods are reported in a way that experiments can be easily reproduced.

-        Add relevant references to the section of curcumin loading techniques

A: Loading was performed according to our laboratory protocols and therefore we do not have references to cite. We have double-checked the sections for all the necessary information for the reproducibiliy of the experiments.

-        The experiment of curcumin release from the nanovesicles is missing and should be included.

A: We have not performed experiments on the release of curcumin. In our bioassays, the cells were treated with curcumin loaded PDNVs without lysis (without release) and we have used free curcumin as control sample. Our aim was to investigate if PDNVs protected curcumin would have an enhanced anti-inflammatory effect respect to free curcumin. 

-        Why in the MTT and trypan blue assays, the groups of NVs, curcumin and curcumin loaded  PDNVs fractions not included to test for their potential in situ toxicity?

A: Regarding curcumin, many studies previously reported that curcumin alone or loaded into EVs is not toxic in the range of concentrations tested (Sinjari et al. Front. Physiol. 10:633.doi: 10.3389/fphys.2019.00633 ). Regarding the crude NVs, we have performed experiments but in the final manuscript we preferred to focus the attention only on the separated fractions as the bioassays have been performed with them.

-        Figure 4 is very low in resolution and should be enhanced

A: We improved the resolution and the distribution of the elements in this figure

-        Table 2 needs adjustment since the texting of the sample and loading methods are in the same level

A: We have tried to improve this aspect.

-        Fig 8 depicted different results for LPS (10 ng/ml) and fr 8 (5 µg) than those presented in Fig 6 b for IL-1β although they are the same experimental settings. why is that?

A: These differences are within the experimental variability we usually find when using biological models such as cellular ones. It is quite common that cells adapt to a stimulus (in our case LPS) elaborating an expression response characterized by different amplitude, the reasons for these differences have to be searched in the number of cell passages, cell confluence, metabolic state, etc.; all these factors can be minimized but not eliminated. For all these reasons, it is crucial to introduce negative and positive controls in our experimental setup, control and LPS condition, respectively. Indeed, in our case, the biological activity of PDNVs has been evaluated by comparing IL-1β expression levels in the experimental conditions (Fr.5, Fr.8 and Fr.9, etc.) versus LPS-induced expression measured for every experiment.

-        Remove the reference from line 663 since the paragraph is about this study

A: References have been removed and the sentence was rephrased.

-        Why is the sonication method of curcumin associated with the highest anti-inflammatory response although it does not have the highest EE% or DL%?

A: We think that the curcumin loaded PDNVs produced by sonication to could be slightly different in size or content or integrity. It is now discussed in the discussion paragraph.

-        Assessing the anti-inflammatory response at the mRNA level only is not sufficient to conclude an anti-inflammatory response. Assays at the protein level should be conducted to support this claim.

A: We agree with the Reviewer that monitoring both mRNA and protein expression levels of cytokines allows a more comprehensive evaluation of the inflammatory response; however, similar studies demonstrate good agreement between mRNA and protein expression levels (Zhang et al. Biomaterials 101 (2016) 321e340; Kim et al. Journal of Food Science Vol. 82, Nr. 5, 2017). Moreover, we are not really set to perform ELISA experiments and within the timeframe given for the revision (i.e. 10 days) it was not possible to arrange such a workflow.

-        The discussion closely resembles the introduction. It should be rewritten to provide justification to the observed results and comparison to previous closely relevant ones.

A: The discussion part vas revised in light of the suggestions. Repetitions to what was already described in the introduction were avoided and the paragraph is now more focused on the justification of the results obtained. Also, it contains some hints about method improvement.

-        EE% and DL% for all different forms of Curcumin loaded preparation are too low. In the discussion, possible justification and methods for improvement should be presented.

A: As above. We included this in the revised discussion part.

Thank you again.

Sincerely yours, Gabriella Pocsfalvi

Round 2

Reviewer 2 Report

accept in current format